# Hybrid Nanomat: Copolymer Template CdSe Quantum Dots In Situ Stabilized and Immobilized within Nanofiber Matrix

**DOI:** 10.3390/nano13040630

**Published:** 2023-02-05

**Authors:** Viraj P. Nirwan, Magdalena Lasak, Karol Ciepluch, Amir Fahmi

**Affiliations:** 1Faculty of Technology and Bionics, Rhine-Waal University of Applied Science, Marie-Curie-Straβe 1, 47533 Kleve, Germany; 2Division of Medical Biology, Jan Kochanowski University in Kielce, Uniwersytecka Street 7, 25-406 Kielce, Poland

**Keywords:** polymer-stabilized quantum dots, hybrid structured nanomaterials, copolymer, electrospinning, non-wovens, fluorescence, biosensors

## Abstract

Fabrication and characterization of hybrid nanomats containing quantum dots can play a prominent role in the development of advanced biosensors and bio-based semiconductors. Owing to their size-dependent properties and controlled nanostructures, quantum dots (QDs) exhibit distinct optical and electronic characteristics. However, QDs include heavy metals and often require stabilizing agents which are toxic for biological applications. Here, to mitigate the use of toxic ligands, cadmium selenide quantum dots (CdSe QDs) were synthesized in situ with polyvinylpyrrolidone (PVP) at room temperature. The addition of PVP polymer provided size regulation, stability, and control over size distribution of CdSe QDs. The characterization of the optical properties of the CdSe QDs was performed using fluorescence and ultraviolet–visible (UV-Vis) spectroscopy. CdSe QDs exhibited a typical absorbance peak at 280 nm and a photoluminescence emission peak at 580 nm. Transmission electron microscopy (TEM) micrographs demonstrated that CdSe QDs having an average size of 6 ± 4 nm were obtained via wet chemistry method. CdSe QDs were immobilized in a blend of PVP and poly(L-lactide-co-ε-caprolactone) (PL-b-CL) copolymer that was electrospun to produce nanofibers. Scanning electron microscopy (SEM), thermal analyses and attenuated total reflectance Fourier-transform infrared spectroscopy (ATR-FTIR) were used to characterize properties of fabricated nanofibers. Both pristine and hybrid nanofibers possessed cylindrical geometry and rough surface features, facilitating increased surface area. Infrared absorption spectra showed a slight shift in absorbance peaks due to interaction of PVP-coated CdSe QDs and nanofiber matrix. The presence of CdSe QDs influenced the fiber diameter and their thermal stability. Further, in vitro biological analyses of hybrid nanofibers showed promising antibacterial effect and decline in cancer cell viability. This study offers a simple approach to obtain hybrid nanomats immobilized with size-controlled PVP-coated CdSe QDs, which have potential applications as biosensors and antibacterial and anticancer cell agents.

## 1. Introduction

Nanostructured organic/inorganic semiconductor hybrids possess exceptional size-dependent optical [1], electrical [2], and magnetic properties [3]. These materials have been explored substantially for biosensing [4,5], photocatalysis [6,7] and photovoltaic [8,9,10] applications. Hybrid nanomaterials based on polymer-templated quantum dots are innovative functional materials, which combine the flexibility and the processability of polymers together with the unique optical collective properties of QDs [11]. Inorganic QDs comprise semiconductor nanoparticles based on metals from the II–VI groups of the periodic table, with an average size between 2 and 20 nm. Favorably, quantum dots offer the possibility of tuning optical properties based on their shape, size, and composition [12,13]. Moreover, templating the QDs within the biopolymers can extend their solubility and compatibility in wide range of biological media in addition to providing control over their morphology. Additionally, inorganic/organic hybrid combinations at nanoscales offer efficient charge separation at the interface between the dielectric shell and semi-conductor core, therefore resulting in magnification of quantum effect and power translation efficiency in addition to improving the photostability. CdSe QDs are remarkable as they offer the possibility of the systematic tuning of the absorption and emission wavelengths as well as wide wavelengths of photoluminescence and stability [14,15,16]. Due to these characteristics, CdSe QDs are preferred for applications such as biomarkers and sensors [17]. CdSe QDs, similar to other inorganic/organic hybrids, consist of a luminescent active core and stabilizing ligands as a shell. Traditionally, the role of stabilizing QDs has been played by ligands such as trioctylphosphine oxide (CT) and octadecylamine (CA) [18]. These ligands are often toxic for biological systems, hence restricting their usage in biological applications [19]. Therefore, combinations such as P3HT-coated CdSe QDs are highly demanded to overcome such issues. Here, P3HT, as an electron donor, is well-suited to coat CdSe QDs as well as enable modification of emissions over a wide wavelength of spectrum [20,21,22]. Similarly, by using polymers as stabilizing agents, it is possible to mitigate the use of toxic ligands while providing stable QDs for biological applications [13,17,23]. In particular, applying the in situ approach restricts agglomeration and facilitates a greater control over the size and the size distribution of QDs [22,24]. Additionally, use of polymer offers a selection of biocompatible materials [13]. Moreover, the presence of functional polymeric matrix can open energy transfer pathways and prevent the self-quenching of the fluorescence of QDs [22,25] and subsequently enhance the efficiency and effectiveness of QDs by structuring them at the nanoscale. Herein, the fabrication of nanofibers via electrospinning offered as an excellent tool for homogenous immobilization of QDs in low dimensional structure. Utilization of nanofibers for structuring hybrid QDs offers two main advantages:a.They provide QDs access to a large surface area due to their characteristic high aspect ratio [26].b.Secondly, they offer QDs a shielding effect, which in turn mitigates toxicity and leaching issues as well as facilitating access to a targeted delivery mechanism [8].

Electrospinning offers a cost effective, relatively controlled mechanism to obtain CdSe QDs structured in nanofibers, which are made from biocompatible polymers [27,28,29]. Hence, in this two-stage investigation, initially, the synthesis of CdSe QDs was performed in situ with PVP polymer to control size and size distribution of QDs. Thereafter, QDs were immobilized in an electrospun nanofibers matrix using a biodegradable and biocompatible copolymer of poly(L-lactide-co-ε-caprolactone) (PL-b-CL). PVP polymer was used as co-spinning agent to stabilize the electrospinning process. Additionally, it facilitated an interface between the PVP-stabilized QDs and the nanofiber matrix. Therefore, hybrid nanofibers with a homogenous distribution of CdSe QDs in nanofiber matrix were fabricated. Finally, the properties of the obtained nanomaterials were characterized using various physicochemical techniques and electron microscopy. Biological analysis was performed to describe antimicrobial activity of nanofibers and their cytotoxic effect on eukaryotic cells. Nanostructured hybrid materials fabricated using this study exhibit suitability for biosensing, biomarker, semiconductor, and photocatalysis applications.

## 2. Materials and Methods

### 2.1. Materials

Cd acetate, selenium powder, PVP (Mw-250 kg) and analytical-grade solvents ethanol/chloroform were purchased from Carl Roth, Karlsruhe, Germany, while NaBH_4_ to obtain NaHSe in ethanol suspension was obtained from Sigma Aldrich. Commercial copolymer of L-lactide and ε-Caprolactone (70:30) (PL-b-CL) (average Mw-200 kg) was purchased from Purasorb^®^, Corbion, The Netherlands.

### 2.2. Methods

#### 2.2.1. Synthesis of CdSe QDs Stabilized with PVP

PVP-stabilized CdSe QDs were synthesized via an in situ approach based on a composition demonstrated in Fahmi et al. [30]. Initially, 0.01 M Cd acetate was dissolved in polymer solution comprising of 0.07 g PVP in ethanol. The mixture was stirred vigorously under N_2_ atmosphere at 40 °C. Simultaneously, 0.01 M Se powder was stirred and suspended in ethanol under constant flow of N_2_. After 1 hour, 0.3 M NaBH_4_ was added to the solution to obtain colorless solution. As soon as the NaHSe suspension turned colorless, it was added dropwise to Cd acetate solution, which gradually turned yellowish, to give CdSe QDs coated with PVP as proposed in **Figure 1**. Solution was left for stirring for 2 h at 40 °C. Resulting CdSe QDs were analyzed using TEM, DLS, and fluorescence/UV-Vis spectroscopy. It was observed that ethanol, which was used as solvent for synthesis QDs, leads to phase separation of co-polymer solubilized in chloroform. Therefore, to prevent phase separation, CdSe QDs were re-dispersed in chloroform before being added to electrospinning solution.

#### 2.2.2. Fabrication of the Nanofibers Immobilized with CdSe QDs

Hybrid nanofibers immobilized with CdSe QDs were prepared by mixing 2 polymer solutions containing 0.85 g PL-b-CL (L-lactide/caprolactone copolymer) in chloroform and 0.21 g PVP (polyvinylpyrrolidone) with CdSe QDs in chloroform. The ratio of weights of PVP and PL-b-CL in electrospinning solution was 4:1. Briefly, 0.21 g PVP was added to 8 mL of PVP-capped CdSe QDs, which were suspended in chloroform in previous stage. Here, PVP was utilized as co-spinning agent to stabilize the electrospinning process and provide better yield. In a separate vial, 0.85 g PL-b-CL was dissolved in 8 mL of chloroform. Then, both solutions were combined and stirred for 24 h at room temperature. After 24 h, the obtained hybrid solution was added into a syringe (5 mL) connected to a capillary (0.8 mm) via PTFE tube. The electrospinning process was carried out with optimized parameters, as presented in Table 1. Nanofibers were collected on the rotating collector covered with non-stick sheet. Similar procedure was followed to obtain pristine PL-b-CL/PVP nanofibers excluding CdSe QDs.

### 2.3. Characterization of Morphology and Size

#### 2.3.1. Morphology and Size Distribution

The morphology of nanofibers was determined using a scanning electron microscope (JSM-IT 100 InTouchScope™, Freising, Germany) at an accelerating voltage of 15 kV. TEM microscopy (JEOL 2200 fs (HR-TEM) and JEOL-10; JEOL Ltd., Tokyo, Japan) was used to provide higher magnification micrographs of PVP-stabilized QDs. The QDs in PBS solution were placed on the carbon surface 200-mesh copper grid (Ted Pella., Redding, CA, USA) for 30 min and drained with blotting paper, while HR-TEM was performed on CdSe QDs suspended in ethanol using JEOL 2200 fs (HR-TEM). ImageJ^®^ [31] was used to analyze SEM and TEM micrographs. Statistical analysis was performed using OriginLab software.

The hydrodynamic diameter of QDs was measured using dynamic light scattering (DLS) in a photon correlation spectrometer (Anton Paar Litesizer 500, Graz, Austria). The refraction factor was assumed to be 1.33, while the detection angles were 15°, 90° and 175°, and the wavelength was 658 nm. PBS was used as a solvent. The data was analyzed using Anton Paar software.

#### 2.3.2. Thermal Analysis

For thermal analysis of fibers, differential scanning calorimetry (DSC) and thermogravimetric analysis (TGA) (Perkin Elmer, Waltham, MA, USA) were used. Briefly, ~10 mg of solid nanofibers samples sealed in an aluminum pan were subjected to multiple heating cycles at the rate of 5 °C min^−1^ from −60 to 200 °C in a chamber flushed with nitrogen. For TGA, ~10 mg of solid fibers were weighed in a ceramic cuvette and heated at 5 °C min^−1^ from 30 °C to 700 °C at nitrogen flow rate of 20 mL.min^−1^.

#### 2.3.3. UV-Vis and Fluorescence Spectroscopy

Before performing spectroscopic analysis, synthesized CdSe QDs were diluted by a factor of 100. Perkin Elmer λ-25 spectrometer was used to measure peak absorbance wavelength exhibited by PVP-coated CdSe QDs. Thereafter, peak absorbance wavelength obtained from UV-Vis analysis was used as excitation wavelength in fluorescence spectrometer. The emission wavelength and photoluminance intensity of PVP-coated CdSe QDs were observed.

#### 2.3.4. Fourier Transform Infrared Spectroscopy (ATR-FTIR)

PerkinElmer Spectrum 2000 spectrometer with an ATR assembly was used to compare changes in specific vibrational frequencies of pristine fibers, and fibers were immobilized with CdSe QDs under infrared at scanning resolution of 2 cm^−1^. Briefly, electrospun nanofibers were cut to ~0.5 cm circle and pressed against crystal on top plate for analysis.

### 2.4. Biological Properties of Nanofibers

#### 2.4.1. Antimicrobial Activity of Nanofibers

The antibacterial activity of the studied nanofibers was tested against *P. aeruginosa* (PAO1). For this purpose, the bacteria were seeded on a 96-well plate and then treated with nanofibers (size 6 mm). Bacterial growth was assessed spectrophotometrically using an optical density test (OD600) in addition to measuring the level of pyocyanin and pyoverdine production by bacterial cells treated with nanofibers. The pyocyanin level was measured at 691 nm. The pyoverdine fluorescence intensity was measured at Ex/Em = 405/460 nm. Experiments were performed in triplicate and results are presented as percentage of control.

#### 2.4.2. Cytotoxic Effect of Nanofibers on Eukaryotic Cells

Two cell lines were used in this study. Human bronchial epithelial BEAS-2B cells (ATCC CRL-9609) were cultured in LHC-9 medium (Gibco). Human lung carcinoma A549 (CCL-185) cells were cultured in F-12K medium (Corning) supplemented with 10% fetal bovine serum and 1% penicillin/streptomycin (100x antibiotic antimycotic solution, Corning). Cells were cultured at 37 °C in a humidified atmosphere and 5% CO_2_. The culture medium was changed every 2 days. The level of surviving nanofiber-treated cells was assessed using the MTS Cell Proliferation Assay Kit (Colorimetric) (abcam). Briefly, cells were seeded in 96-well plates and cultured for 24 h to reach the appropriate cell confluence. Next, the cells were treated for 24 h with UV-sterilized nanofibers (sterilized for 30 min). It has been reported that exposing PVP polymer to UV light (254 nm) induces crosslinking. However, the crosslinking was reported in solution phase and in the presence of crosslinking agents. Here, no noticeable crosslinking was observed as the exposure time was insufficient. Nanofibers (size 35 mm) were immobilized with inserts to achieve the appropriate cell/nanofiber contact. Untreated cells were used as control. After 24 h incubation, the nanofibers were removed, and the MTS reagent was used according to the manufacturer’s instructions.

#### 2.4.3. Inhibition of Proliferation of Cancer Cells

The A549 cells were seeded in 6-well plates and cultured for 72 h in the presence of tested UV-sterilized nanofibers (size 35 mm). Next, the nanofibers were removed, and cells were observed by optical microscopy (×64).

## 3. Results and Discussion

One route to obtain stable QDs colloids is to attach a large ligand, which can maintain QDs in a suspended state due to steric hindrance. Here, PVP was as an excellent candidate to be used as ligand due to its biocompatible nature and presence of a long polymer chain, which provided hindrance effect. Hence, synthesis of CdSe QDs was performed in situ with PVP and resulted in neon yellow solution, which was highly stable. Their morphological and spectroscopic properties were analyzed using TEM/DLS and fluorescence/UV-Vis spectroscopy. The TEM micrographs showed that the QDs were coated with a thin layer of PVP polymer (Figure 1). The core diameter of PVP-coated CdSe was measured to be between 5–15 nm with a mean of 6 ± 4 nm, and their average hydrodynamic diameter was 23 ± 7 nm. DLS data demonstrated that the synthesized QDs had fewer aggregates, possessing a polydispersity index PDI −0.27 (Figure 2). Indeed, the layer of PVP around QDs had resulted in stabilization and minimized agglomeration. Further, a typical CdSe QDs diffraction pattern was observed from the HR TEM showing the crystal structure of CdSe QDs to the third order.

As seen in Figure 3, CdSe QDs showed a typical UV-visible spectroscopy absorption behavior. CdSe QDs stabilized with PVP had a relatively narrow absorption peak at 280 nm, with an absorption range from 320 to 240 nm, while no absorption peak was observed for PVP polymer solution, which was used as control.

PVP-stabilized CdSe QDs showed a strong photoluminescent behavior, as seen in Figure 3. Here, QDs were excited using wavelength of 290 nm. PVP-stabilized QDs showed emission wavelength at 580 nm. Photoluminescence spectroscopy shows that CdSe QDs coated with PVP have favorable fluorescent characteristics, and they can be utilized potentially as biomarkers. For instance, functionalizing the surface of CdSe QDs with glycoproteins and utilizing their luminescence properties has been demonstrated to be a simple, cost-effective and specific method to detect α-fetoproteins [32]. Here, using PVP in situ as a stabilizing and coating agent allowed a greater control over size and size distribution of CdSe QDs. Moreover, it prevented the use of surfactants, which are used as stabilizing agents and are shown to be toxic for biological applications [33]. Further, it can potentially help mask the toxicity of the QDs towards biological systems or influence the uptake of CdSe [34].

Recognizing this potential of the CdSe QDs, they were immobilized in a PVP/PL-b-CL nanofiber matrix. Electrospinning was used to fabricate hybrid nanofibers containing homogenously distributed PVP-stabilized CdSe QDs. Electrospinning of PL-b-CL alone was highly unstable as chloroform was required to dissolve co-polymer to obtain the electrospinning solution. Use of chloroform favored rapid evaporation at the spinneret, causing erratic jet ejection and blockage due to solidification of polymer. Hence, PVP was used to blend PL-b-CL copolymer for two reasons. Firstly, being used to stabilize CdSe QDs, it could facilitate stronger interactions among CdSe QDs and the fiber matrix. Secondly, its high molecular weight grants easy electrospinnability in addition to compatibility with the solvent, promising a homogenous blend. It was noticed that the addition of PVP as co-electrospin agent had a moderate effect on the stability of the electrospinning process. After electrospinning, fibers were obtained with an intermediate yield possessing cylindrical morphology. SEM was used to study the morphology of pristine and CdSe QDs-immobilized nanofibers, which provided micrographs, as seen in Figure 4. Using ImageJ, micrographs were analyzed to obtain information about the nanofibers’ diameter. Statistical analysis of the data revealed that pristine fibers possessed a mean diameter of 2 µm, which was twice the mean diameter of the immobilized fibers at 1 µm. It is presumed that immobilization with metallic nanoparticles provided relative stability to electrospinning process, and the presence of QDs led to a change in conductivity, viscosity, and surface tension [35,36]. Interestingly, both pristine and functionalized nanofibers possessed rough surface features with dimensions below 500 nm. It is a consequence of electrospinning at high humidity and low temperature to prevent rapid evaporation of solvent at the spinneret. Electrospinning in the presence of high humidity during results in vapor-induced phase separation where during the jet formation water vapor is absorbed in the jet. Absorption of water vapor leads to phase separation in hydrophobic polymers, creating polymer-rich and polymer-deficient regions. When the highly volatile solvent (in this case chloroform) evaporates quickly, it locks the phase geometry on deposition providing porous/rough surface features, as seen in the inset of Figure 4a,b [37,38]. Such remarkable morphology can potentially be helpful for cell adhesion, and enhanced surface area can increase the surface activity of scaffolds and the effectivity of QDs [39,40]. The same process also results in high diameter distribution presumably due to the combination of highly volatile solvent, high humidity and hydrophobic polymer [38]. Due to high volatility and rapid evaporation of solvent, the solidification of the polymer was encountered at the spinneret tip, resulting in irregularity during the electrospinning process. Hence, the electrospinning was performed at low temperatures to minimize evaporation and stabilize the process. A statistical summary of the nanofiber dimensions can be seen in Table 2.

Nanofibers were collected as a non-woven mat and characterized with ATR-FTIR, giving spectra as seen in Figure 5. The ATR-FTIR analysis shows a slight difference between absorption spectra recorded from pristine and CdSe QDs-immobilized nanofibers. The stretching vibration of the O-H group due to absorbed H_2_O molecules through PVP could be seen at 3454 cm^−1^ in the pristine nanofibers sample. A similar peak at 3439 cm^−1^ was observed for fibers containing PVP-capped CdSe QDs [41,42]. Here, utilization of PVP as stabilizing agent has potentially facilitated hydrogen bonding between PVP, capping CdSe QDs, and copolymer present in the fiber matrix (Figure 2). In both samples, overlapping asymmetric stretching of C-H (CH_2_ and CH_3_) was represented at 2945 cm^−1^ [43]. Further, at 1755 cm^−1^, a peak due to stretching of ester C=O corresponding to copolymer was noticed [44], while C-O stretching in PVP was noticeable at 1660 cm^−1^. Absorption peaks at 1454 cm^−1^ and 1382 cm^−1^ correspond to asymmetric and symmetric C-H bending, while absoprtion at 1272 cm^−1^ represents CH_2_ wagging. Asymmetric and symmetric C-O-C stretching were also present at 1181 cm^−1^ and 1129 cm^−1^, respectively [43,45]. C-C breathing was restricted in samples with CdSe QDs/PVP, presumably due to its interaction with PVP and PL-b-CL/PVP matrix, shifting the peak from 754 cm^−1^ in pristine samples to 736 cm^−1^ in functionalized fibers [46]. Additionally, N-CO bends between 650 and 533 cm^−1^ were similar in both samples [46,47]. Finally, the characteristic fingerprint region observed in FTIR analysis of polymers used for fabrication of nanofibers and capping CdSe QDs remained unchanged before and after inclusion of CdSe QDs. Overall, immobilization of CdSe QDs stabilized with PVP promoted weak interactions among inorganic and organic moieties. This promoted a homogenous distribution of QDs in fibers in addition to helping to minimize potential leaching of inorganic materials in biological media.

The characterization of thermal properties is relevant as immobilizing the nanofibers with QDs can affect its degradation behavior and thermal stability. Additionally, both the electrospinning process and the functionalization of nanofibers using QDs can influence the crystallinity of the semi-crystalline PL-b-CL/PVP copolymer blend. Moreover, solvent, processing temperature and humidity too can play an important role in defining the crystalline properties of both pristine and functionalized nanofibers. Initially, thermogravimetric analysis was employed to characterize the degradation behavior of pristine and functionalized nanofibers, followed by the analysis of phase change properties of nanofibers and their variations in the presence of QDs using differential scanning calorimetry. The thermogram seen in Figure 6 shows the degradation of behavior of nanofibers when subjected to a temperature program from 30–700 °C. Major findings of the TGA analysis are highlighted in Table 3. Both nanofiber samples exhibited a three-step degradation profile, implying an electrospinning formulation consisting of multiple constituents such as PL-b-CL/PVP. There was no weight degradation observed until 280 °C, which was recorded as the onset temperature for immobilized nanofibers. Compared to those, the pristine nanofibers had a relatively higher onset temperature of 304 °C. This indicated that the QDs structured inside the nanofiber matrix acted as heat spots leading to a decrease in onset temperature.

For pristine nanofibers, the maximum mass degradation rate occurred at 338 °C. This degradation step corresponds to the melting and breakdown of intermingled segments. A similar step was observed at 337 °C in the thermogram of functionalized nanofibers. Further, a second degradation event occurred at 380 °C for pristine fibers and 370 °C for functionalized fibers, which corresponds to the degradation of low crystalline PCL of the PL-b-CL copolymer.

Subsequently, the end of the degradation temperature for pristine nanofibers was observed at 384 °C, where presumably most of the PL-b-CL copolymer was decomposed. Interestingly, a third degradation step was noticed at 431 °C, resulting from the condensation and cyclization of the polyromantic structures of PVP. Identical peak was noticed for functionalized nanofibers where a similar third degradation step occurs at 434 °C. However, due to the presence of QDs in functionalized nanofibers, their end of degradation temperature was delayed until 429 °C. After the end of temperature program the pristine nanofibers showed a total weight loss of 98%, whereas the total weight loss for nanofibers functionalized with CdSe QDs was significantly reduced at 91% loss [48,49].

Further analysis of thermal characteristics and differences in phase change behavior were observed using DSC. Analysis of nanofiber samples using the DSC temperature program from −0 to 200 °C showed a relatively complicated behavior with multiple melting and crystallization peaks (Figure 7). The samples were subjected to several heating cycles, which had impact on their phase change temperatures. As expected, the first heating cycle revealed the rapid setting of polymer during electrospinning. The DSC thermogram revealed a glass transition peak (T_g_) at 35 °C and 45 °C for pristine and functionalized fibers, respectively. A crystallization peak was observed at 65 °C and 70 °C for pristine and functionalized fibers, respectively. Cold crystallization peaks could be observed in pristine nanofibers samples at 140 °C. However, a similar phase change event is absent for functionalized nanofibers. Pristine PL-b-CL/PVP nanofibers exhibited two distinct melting peaks at 156 °C and 184 °C [50]. In contrast, CdSe QDs-immobilized nanofibers displayed a broad melting curve with two adjacent peaks with slight overlap 133 °C and 158 °C. As the samples went through the second heating cycle, their response was smoother with fewer phase change events. Noticeably, T_g_ was absent in thermogram of both samples. Here, it seems that PCL exhibited some degree of crystallinity with an individual characteristic endotherm peak around 40 °C, which was absent in the first heating cycle and is lower compared to PCL homopolymer [44]. A weak crystallization peak could still be observed in pristine nanofiber samples at 63 °C. In contrast, CdSe QDs-immobilized fibers had a sharp crystallization peak at 80 °C. In the second heating cycle, both pristine and CdSe QDs-functionalized fibers showed a single melting event occurring at 156 °C and 166 °C for pristine and CdSe QDs-functionalized fibers, respectively. Overall, the inclusion of CdSe QDs in the PL-b-CL/PVP nanofiber system increased their amorphosity. As expected, the PVP component in the nanofibers had no peaks largely due to its amorphous nature [51]. As observed from the first heating cycle, the presence of structured CdSe QDs within the matrix function as heating spots bringing the melting endotherm peak below by a few degrees compared to the pristine nanofiber samples. As observed in the second heating cycle, the occurrence of melting event at a slightly higher temperature indicates a loss of structure and change in distribution of CdSe QDs after the first heating cycle.

The results of the antibacterial test showed that the nanofibers inhibited the growth of bacteria. As shown in Figure 8A (OD600), both the pristine PL-b-CL/PVP nanofiber excluding QDs and immobilized with CdSe QDs inhibited bacterial growth by ~36% and ~48%, respectively, compared to the control. The antibacterial effect is also confirmed by the results of measuring the level of two pigments produced by *P. aeruginosa*—pyocyanin and pyoverdine (Figure 8B,C, respectively). In the presence of both types of nanofibers, a decrease in the production of dyes was observed, especially for bacteria treated with CdSe QDs nanofibers. Thus, nanofibers immobilized with CdSe QDs are slightly more toxic to *P. aeruginosa* than pristine PL-b-CL/PVP nanofibers. These results are consistent with the observed change in the color of the bacterial culture media (Figure 8D). In the case of the untreated culture of *P. aeruginosa* (control), the medium was light green, which indicates the production of the primary dye and virulence factor—pyocyanin. For bacterial cells treated with nanofibers, the medium was light brown, which may indicate the blockage of the basic metabolic pathways of *P. aeruginosa* and thus explain the reduced growth of bacteria treated with nanofibers [52].

The non-cancer cells (BEAS 2B) and cancer cells (A549) were treated with tested nanofiber for 24 h, and their cytotoxicity potentials were evaluated toward those cells employing MTS assay. Figure 8E illustrates the decrease of cell viability compared to control. There is no significant decrease in samples treated with PL-b-CL/PVP nanofibers. However, in presence of PL-b-CL/PVP + CdSe QDs the decrease of cell viability to 20% was observed. Similar results were observed for cancer cells (Figure 8F) A549. The significance of cytotoxicity in shown for PL-b-CL/PVP + CdSe QDs. The cell growth inhibition of studied nanofibers against A549 cells is shown in Figure 8G. No inhibition in cell growth was observed in the control (untreated cells) after 3 days. In the presence of PL-b-CL/+PVP nanofiber, one can observe gradual growth inhibition. In the case of PL-b-CL/PVP CdSe QDs, pronounced growth inhibition was present. The obtained results indicate that loading CdSe QDs into the PL-b-CL/PVP nanofibers improves its inhibitory effects. The explanation for this is the inhibitory effect of CdSe quantum dots (QD) on cancer cells, which is very often associated with blocking many signaling pathways, e.g., Rho-associated kinase (ROCK) activity [53], or can cause ultrastructural changes in cells, such as organelle degeneration, chromatin condensation, and aggregation [54].

Here, these forms of hybrid PL-b-CL/PVP CdSe QDs nanofibers can be used as antimicrobial materials for wound infections or anticancer materials acting locally (e.g., in cancer treatment with photodynamic therapy). In addition, hybrid PL-b-CL/PVP CdSe QDs nanofibers can be used as a platform with further modifications as a biosensor for the detection of various biomacromolecules in biological fluids.

## 4. Conclusions

A simple concept to template CdSe QDs via in situ approach using polyvinylpyrrolidone (PVP) has been explored successfully. Use of PVP as stabilizing agents gave a higher degree of control over the size of QDs and provided exceptionally stable CdSe QDs colloid. As observed through TEM and DLS measurements, PVP-stabilized CdSe QDs had a narrow size distribution. In situ synthesized CdSe QDs exhibited a sharp photoluminescence peak at 580 nm. PVP-stabilized CdSe QDs were immobilized in the matrix of electrospun PL-b-CL/PVP nanofibers. Here, interaction between stabilizing agent, and nanofibers matrix and its effect on physicochemical properties of the nanofibers were investigated. Overall, obtained fibers were uniform cylindrical and free of beads. The fibers were decorated with rough surface features, enhancing the exposed surface area promoting QDs activity. The presence of PVP as stabilizing agents provided increased coordination domains, which delivered homogenous distribution of CdSe QDs, and stability in nanofiber matrix. Furthermore, CdSe QDs immobilized fibers had better thermal stability, where QDs functioned as heating spots. Antimicrobial assays clearly indicate the antibacterial effect of CdSe QDs-functionalized nanofibers against drug-resistant *P. aeruginosa* PAO1. Finally, testing of pristine and functionalized scaffolds via in vitro viability assay using BEAS-2B and A549 cells showed a favorable response as compared to the control. Moreover, growth inhibition of cancer cells in the presence of functionalized nanofibers is observed. These results demonstrate the potential of CdSe QDs functionalized nanofibers for use in various biological applications, including biosensors and antibacterial, anticancer cell agents. The fine-tuning of the electrospinning process along with additional biological analysis of fibers will further improve and establish the potential of nanofibers in medicinal nanotechnology and diagnostics.

## Data Availability

Not applicable.

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
