# Peer review of "Hybrid Nanomat: Copolymer Template CdSe Quantum Dots In Situ Stabilized and Immobilized within Nanofiber Matrix"

_nanomaterials, 2023, doi:10.3390/nano13040630_

Round 1
Reviewer 1 Report
There were some reports on the CdSe quantum dots immobilized within nanofiber matrix, a though this part in the manuscript is not novel, however, there is a certain reference value for its use in the antibacterial test. In addition, there are the following problems:
1. Although the lattice fringes and diffraction patterns were shown in Fig. 1, which only proved that crystals are formed, it cannot prove that they are CdSe quantum dots.
2. It is well known that trace amounts of Cd+ are highly toxic substances. How can been demonstrate that there is no Cd+ ion in the nanofibers and hybrid solution? This is a key problem. It is better to have ICP analysis results.
3. Will the spectral properties of CdSe QDs change before and after Electrospinning? It is better to give test results.
4. The FTIR analysis cannot provide the relevant results of CdSe QDs. The analysis in the manuscript is of little significance because the hydrocarbon has various vibration peaks in the near infrared. The information of CdSe QDs can be obtained by micro Raman instrument.
Reviewer 2 Report
This manuscript reports the preparation of CdSe QD-functionalized PL-b-CL-PVP nanofibers, and their corresponding characteristics were also investigated using FTIR, UV, PL, and TEM. The results of this article present significant scientific contributions with respect to the development of biomedical. I feel that major modifications are necessary before publication can be considered.
(1) The biggest concern of the reviewer is the molecular interaction of CdSe QD-PVP and PL-b-CL. The authors need to provide the schematic representation of hydrogen bonding between them. In line 243-244, they have mentioned that “Presence of PVP as stabilizing agent allowed hydrogen bonding between QDs and fibers matrix”. Because the authors also said that “Immobilization via QDs provided relative stability to electrospinning process, which lead to smaller mean diameter and diameter distribution” in line 219.
Besides, in section 2.2.2, what does PVP K90 mean?
(2) The second concern of reviewers is why the peaks of the absorption and fluorescence spectra do not partially overlap in Figure 3? In addition, Figure 3 needs to add arrows to indicate the above absorption and fluorescence data.
(3) In the abstract part, it is necessary to write some representative data or related content.
(4) In Table 1, the used amount of PVP and PL-b-CL is 2.66 and 10.625%, respectively. However, how to get the 2.66% of PVP from the CdSe QD-PVP?
(5) In Figure 1c, the detail description of diffraction pattern is missing.
(6) In Figure 2, the solvent in DLS should be added.
(7) In line 200, the statement of “it is clear through the analysis that CdSe QDs stabilized with PVP have favorable fluorescent characteristics to be utilized as biomarker. Moreover, using PVP as stabilizing agent mitigate the use of toxic ligands, which are often used as stabilizing agents” was unclear. The author should explain clearly in the revised manuscript.
(8) In SEM analysis, what’s the reason that the nanofibers’ diameter revealed that pristine fibers possessed a mean diameter of 2µm, which was almost twice the mean diameter of immobilized fibers at 1µm?
(9) The reason of “pristine fiber possessed nanopores on the surface” should be given in the revised manuscript.
(10) PVP polymer itself does not have OH groups, why does Figure 5 have this wavenumber of 3454 cm-1? Besides, the functional peaks (i.e., C=O, C-O-C) and wavenumber should be provided in Figure 5.
(11) Figure 6 needs to add arrows to indicate the weight and derivative weight.
(12) Figure 7 is too cluttered to be easily recognized. Where is the 1st scan? In addition, Tg, Tc and Tm should be indicated in Figure 7.
(13) In Figure 8, the concentration of polymer should be given. Also, the scale bar of Figure 8D is missing.
(14) In section 2.1, the molecular weight of PL-b-CL should be given.
(15) In line 16, it should be analyzed, not analysis.
(16) In line 299, it should be 200oC, not ‒200oC.
Reviewer 3 Report
First of all the manuscript contains plenty of grammatical errors. It should be read by a native speaker or senior scientist with good scientific/technical vocabulary.
Authors claim that the prepared composite nanomaterial is suitable for bio applications. How the heavy metal Cadmium along with toxic Selenium can be non-toxic in the form of QD, and how water-soluble PVP can lower their toxicity? Please explain your point of view
Abstract – the last sentence has no connection with the manuscript at all.
Page 2, row 80 – “leading to contamination” – of what?
Page 2, rows 96-97 – molar concentration is not appropriate for use with polymer solutions due to the wide distribution of the polymers' molecular weight. Here more suitable would be the wt.%. Also somewhere authors use molar concentrations and somewhere %-s – please unify.
Page 3, row 102 – “few hours” – if it is about the precise preparation process – please clarify the exact time.
Why the prepared QDs were re-dispersed in CHF? Why not use it in its original form?
Page 3, rows 106-111 – the preparation description is barely understandable. What and why was mixed with water? What and why was mixed in CHF?
Page 3, row 123 – why PBS solution was used for TEM sample preparation? PBS contains salts, which may impact the imaging results.
Page 3, rows 140-141 – the last sentence is not understandable at all.
Next, based on the results of DLS in Fig.2 and TEM micrographs (the quality of which are quite low to provide some reliable information), the size of the prepared particles is about 23 nm, whereas the typical size of QD is 2-10 nm. How authors can confirm that produced particles are QDs? Please explain.
Page 6, row 200 - Based on which properties the synthesized particles may be used as biomarkers? Please explain in detail.
Page 7, rows 216-219 what kind of immobilization and functionalization it is about? Please clarify.
Nanofibers typically have diameters below 1 µm. Fibers produced in this work have a mean diameter above 2 µm, which makes them NOT nanofibers, but more likely microfibers. Please do not call them so, and change the title of the manuscript.
Fig. 4. – based on what authors call the rough surface of the polymer fibers “nanopores”? Nanopores cannot be visible by SEM. Maybe it could be possible by TEM, but definitively not by SEM, especially at low magnification, like the one used for the micrograph in Fig.4b.
If the used solvent – CHF – is too volatile, why authors lowered the electrospinning temperature? Why not add some other good solvent but with a higher boiling point? for example, DMF, which also will improve the solution conductivity.
FTIR spectra – it would be more suitable for the reader to see two overlapped spectra – the differences would be more visible. Also, how the spectra were recorded in the ATR mode? Please describe the sample preparation, at least very briefly.
How can be explained such a dramatic weight loss, if the measurements were performed in nitrogen? PVP should be pyrolyzed and carbon fibers had to be formed.
The authors state, that fibers were sterilized by UV light (I assumed at 254 nm) for cell tests. For how long? Did the authors count the fact, that PVP is crosslinking under UV light? It means that the PVP in the fibers becomes insoluble in the water and could block the QD leaching. In other words, the sample preparation itself may alter the tested material significantly, and the obtained results may be not relevant.
Also, a similar situation may happen to the PL-b-CL. Did the authors study these possibilities?
As was originally expected, the results of the cell/bacteria viability tests confirm significant toxicity of the fibers with CdSe QDs towards not only bacteria and cancer cells, but also non-cancerous ones. As far as I can see, this show limited possibility to use the produced material in biomedical applications. How authors can comment on this? Maybe it has potential in some other particular applications, not evident from the described in the manuscript?
Round 2
Reviewer 2 Report
The authors have revised the manuscript according to the comments and concerns of the reviewers. But, I have two suggestions for this revision.
1. Page 10, the size of atom or functional groups between polymer chain and PVP is not reasonable. The authors need to redraw them.
2. Page 13, the indication of tg, tc, and tm should be Tg, Tc, and Tm.
Author Response
Please see the attaachment

Reviewer 3 Report
Now, after the revision I really can say that authors did a good work. The improvement of the manuscript is clearly evident and it moved the work to another, higher level. Some minor language issues still can be "polished", but the overall impression is good.
Authors responded to all my questions/comments. I still have some concerns about the toxicity of the CdSe QDs and crosslinking of the PVP, this is why I think, that responses to my last 3 questions should be included in the manuscript. This would help other readers better understand the work without raising the same questions as me.
